# Direct RT-qPCR Assay for the Detection of SARS-CoV-2 in Saliva Samples

**DOI:** 10.3390/mps5020025

**Published:** 2022-03-07

**Authors:** Francesco Saverio Tarantini, Siyu Wu, Harry Jenkins, Ana Tellechea Lopez, Hannah Tomlin, Ralph Hyde, Katarzyna Lis-Slimak, Jamie Louise Thompson, Sara Pijuan-Galitó, Danielle Scales, Kazuyo Kaneko, Jayasree Dey, Emily Park, Jack Hill, I-Ning Lee, Lara Doolan, Asta Arendt-Tranholm, Chris Denning, Claire Seedhouse, Andrew V. Benest

**Affiliations:** Asymptomatic Testing Service, Biodiscovery Institute, School of Medicine, University of Nottingham, Nottingham NG7 2RD, UK; francesco.tarantini1@nottingham.ac.uk (F.S.T.); mbxsw6@exmail.nottingham.ac.uk (S.W.); msxhj3@exmail.nottingham.ac.uk (H.J.); paxaate@exmail.nottingham.ac.uk (A.T.L.); hannah.tomlin@nottingham.ac.uk (H.T.); mszrh@exmail.nottingham.ac.uk (R.H.); mszkl@exmail.nottingham.ac.uk (K.L.-S.); jamie.thompson@crick.ac.uk (J.L.T.); sarapijuangalito@gmail.com (S.P.-G.); mbxds5@exmail.nottingham.ac.uk (D.S.); mszkk2@exmail.nottingham.ac.uk (K.K.); stxjd10@exmail.nottingham.ac.uk (J.D.); stxep4@exmail.nottingham.ac.uk (E.P.); stxjh26@exmail.nottingham.ac.uk (J.H.); pazil1@exmail.nottingham.ac.uk (I.-N.L.); mzxld1@exmail.nottingham.ac.uk (L.D.); paxaat@exmail.nottingham.ac.uk (A.A.-T.); plzcnd@exmail.nottingham.ac.uk (C.D.); claire.seedhouse@nottingham.ac.uk (C.S.)

**Keywords:** COVID-19, saliva, qPCR

## Abstract

Since mid-2020 there have been complexities and difficulties in the standardisation and administration of nasopharyngeal swabs. Coupled with the variable and/or poor accuracy of lateral flow devices, this has led to increased societal ‘testing fatigue’ and reduced confidence in test results. Consequently, asymptomatic individuals have developed reluctance towards repeat testing, which remains the best way to monitor COVID-19 cases in the wider population. On the other hand, saliva-based PCR, a non-invasive, highly sensitive, and accurate test suitable for everyone, is gaining momentum as a straightforward and reliable means of detecting SARS-CoV-2 in symptomatic and asymptomatic individuals. Here, we provide an itemised list of the equipment and reagents involved in the process of sample submission, inactivation and analysis, as well as a detailed description of how each of these steps is performed.

## 1. Introduction

Diagnostic tests for COVID-19 routinely rely on the detection of SARS-CoV-2 proteins or nucleic acids in nasopharyngeal samples through lateral flow (LFT) or polymerase chain reaction (PCR), respectively [1]. With 95–99% specificity and sensitivity, PCR approaches, such as reverse transcriptase quantitative PCR (RT-qPCR) or loop-mediated isothermal amplification (LAMP), are undoubtedly the mainstay of COVID-19 tests [2]. Nevertheless, the use of nasopharyngeal swabs as the means of sample collection for these tests presents several limitations, most notably due to failure to reach the correct nasopharynx target site (with consequences on the reproducibility and standardisation of the test), the discomfort accompanying and following the administration of the swab, and not least the necessity of employing an intermediate RNA extraction step. More recently, direct RT-qPCR approaches have been developed that allow for the detection of SARS-CoV-2 nucleic acids in a heat-inactivated saliva sample, thus removing the need for swabbing and bypassing the requirement for RNA extraction [3,4].

Here, we describe in detail a direct RT-qPCR COVID-19 test based on the WHO-approved Charité protocol [5] (https://www.who.int/docs/default-source/coronaviruse/protocol-v2-1.pdf (accessed on 1 June 2021)). In its current version, the assay employs a triplexed combination of DNA primer and probe oligonucleotides (from the Centers for Disease Control and Prevention (CDC) and a Food and Drug Administration (FDA)-approved panel) for the detection of the SARS-CoV-2 nucleocapsid (N) and envelope (E) genes, as well as the human RNase P gene as an internal control. During a typical PCR cycle, each set of primers is used to amplify its target from a cDNA template obtained by the reverse transcription of the viral genome at the beginning of each reaction. N, E, and RNase P probes each contain a reporter fluorophore (FAM, HEX, or ATTO 647, respectively) in steric proximity to a quencher dye and anneal to a specific target sequence within the N, E, or RNase P amplicons. The annealed probes are cleaved during the DNA extension phase by the exonuclease activity of the DNA polymerase, thus freeing the fluorophore from the quencher and resulting in the production of a fluorescent signal detectable around 520 nm (FAM), 556 nm (HEX), or 667 nm (ATTO 647). The level of fluorescence for each reporter is monitored during each PCR cycle and is used to determine the presence of viral particles in the sample tested. The key advantage of this method is that the only step between sample submission and analysis is heat inactivation. Whilst necessary to neutralise any infectious agents present in the saliva (including SARS-CoV-2) [6] and reduce the viscosity of the sample, heat treatment also promotes the release of genetic material from viral particles without the need for the chemical agents and reduces processing times required for nucleic acid extraction and purification. Given the smaller scale of our operation, processivity has never been a limiting factor. However, the specificity and sensitivity demanded of the assay from the UK regulators were hard requirements that we had to meet. In practice, the occurrence of problematic samples is low enough to not have a meaningful impact on the overall speed of the process and largely compensated for by the time saved by not using other treatment options or extracting the RNA.

Notwithstanding its simplicity and rapidity, this approach allows for results with diagnostic sensitivity and specificity of 99.4% and 99.6%, respectively, placing heat-inactivated saliva testing on the same level with (or above) current extracted molecular testing options (https://www.gov.uk/government/publications/assessment-and-procurement-of-coronavirus-covid-19-tests/coronavirus-covid-19-serology-and-viral-detection-testing-uk-procurement-overview (accessed on 24 January 2022). Furthermore, the assay satisfies the technical requirements set by ISO 17025:2017 for general testing and has allowed the Asymptomatic Testing Service at the University of Nottingham to receive accreditation as a COVID test provider by the United Kingdom Accreditation Service (UKAS) in December 2021. Finally, the same procedure can be adapted to analyse pooled samples to increase the testing capacity and optimise the use of time and reagents. However, this comes at the cost of a partial loss in sensitivity, making the pooling strategy best used when the expected prevalence of positive individuals within the tested population is low (5–10%). The preference of pooled versus non-pooled approaches is, therefore, a choice for the testing laboratory based on their stakeholders, customers, and the requirements of any overseeing accreditation agencies.

## 2. Equipment and Reagents

### 2.1. Sample Collection

FluidX 48-format, 1.9 mL, external thread, next-gen jacket, tri-coded cryotubes (Brooks Life Sciences, Manchester, UK, 30128-1)Collection aid (single-use straws or funnels)Leakproof, transparent Ziploc bagLidded container appropriate for transportation of UN 3373 biological substances category BSamples were donated voluntarily by staff and students at the University of Nottingham, UK, in accordance with Faculty of Medical and Health Science Ethical Approval, University of Nottingham. Informed consent was used throughout the process.

### 2.2. Heat-Inactivation

Class I microbiological safety cabinet (MSC I)20 × 20 metal storage rackMultipurpose benchtop oven (Genlab, Widnes, UK, OV/50/TDIG/SS/F)Thermometer with submersible probe (Thermo Fisher, Waltham, MA, USA. 200590998)70% industrial methylated spirits (IMS) for surface decontamination.

### 2.3. Sample Registration

FluidX Perception HD whole rack reader (Brooks Life Sciences, 20-4018)FluidX Intellicode decoding software (Brooks Life Sciences, 20-3503)FluidX 8 × 6 barcoded storage rack for 48-format, 1.9-mL cryotubes (Brooks Life Sciences, 430128-1).

### 2.4. qPCR Setup and Analysis

FluidX IntelliXcap 48-Format Screw Cap Tube Rack Decapper/Capper (Brooks Life Sciences, 46-8011)C1000 Touch thermal cycler with CFX96 optical reaction module for real-time PCR systems (Bio-Rad, Watford, UK. 1841100, 1845097)Bio-Rad CFX Mestro 1.0 data analysis software (version 4.0.2325.0418)Class II microbiological safety cabinet (MSC II)E1-ClipTip electronic adjustable tip spacing multichannel equalizer pipette (Thermo Fisher Scientific, 4672050BT)ClipTip 200 filtered pipette tips (Thermo Scientific, 13286269)Eppendorf twin.tec 96-well PCR plate (Gillingham, UK, EP951020401)Thermo Scientific adhesive PCR plate seals (Thermo Fisher Scientific, AB0558)UltraPlex 1-Step ToughMix, 4X (Quantabio, Beverley, USA. 95166)SARS-Related Coronavirus 2 (SARS-CoV-2) External Run Control (Buffalo, NY, USA Zeptometrix,; NATSARS(CoV2)-ERC)SARS-Related Coronavirus 2 (SARS-CoV-2) Negative Control (Zeptometrix, NATSARS(CoV2)-NEG)CDC nCOV_N2 FWD primer (5′-GACCCCAAAATCAGCGAAAT-3′)CDC nCOV_N2 REV primer (5′-TCTGGTTACTGCCAGTTGAATCTG-3′)CDC nCOV_N2 probe (5′-FAM-ACCCCGCAT/ZEN/TACGTTTGGTGGACC-IBFQ-3′)Charité/Berlin E_sarbeco FWD primer (5′-ACAGGTACGTTAATAGTTAATAGCGT-3′)Charité/Berlin E_sarbeco REV primer (5′-ATATTGCAGCAGTACGCACACA-3′)Charité/Berlin E_sarbeco probe (5′-HEX-ACACTAGCC/ZEN/ATCCTTACTGCGCTTCG-IBFQ-3′)CDC RNase P FWD primer (5′-AGATTTGGACCTGCGAGCG-3′)CDC RNase P REV primer (5′-GAGCGGCTGTCTCCACAAGT-3′)CDC RNase P probe (5′-ATTO647-TTCTGACCT/ZEN/GAAGGCTCTGCGCG-IBFQ-3′)Nuclease-free water.

## 3. Procedures

### 3.1. Sample Submission and Registration. Time for Completion: 1–2 min per Sample

To guarantee optimal results, samples should be submitted before or at least 1–2 h after any meals, drinks, or oral hygiene practices, in order to avoid diluting the saliva or introducing contaminants or inhibitors into the final reaction.Only oral saliva should be provided, as forcefully expelled deep-throat sputum (DTS) may negatively affect the sample transfer steps, leading to sub-optimal results.An adequate amount of saliva should be submitted to ensure that the sample is representative and can withstand the heat inactivation process.Sample should be provided into barcoded, external thread tubes that allow for sample traceability and minimise the risk of spills and cross contamination between sample tubes during processing.A straw (or similar collection aid) should be used to provide the sample within the tube.Filled sample tubes should be bagged in leakproof, transparent bags to avoid spills and allow for easy inspection before processing.Bagged samples tubes can be transported to the testing laboratory in adequate rigid, lidded containers according to UN 3373 specifications.

### 3.2. Heat-Inactivation of Saliva Samples. Time for Completion: 1.5 h (for 400 Samples)

8.Turn on the oven and preheat it to 110 °C.9.Transfer the bagged sample tubes inside the MSC I for inspection.10.Place a sterilised 20 × 20 metal rack inside the cabinet.11.Inspect sample tubes for damage or obvious tampering and leakage prior to unbagging and placing them into the metal rack for the inactivation step. It is recommended to keep one empty slot between tubes in the rack (Figure 1) and to reserve one corner slot for the control tube (a tube with a pierced lid and filled with water). Up to 199 samples can be placed in a single 20 × 20 rack this way. If fewer than 199 samples need to be inactivated skip straight to step 13.
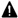
 **CRITICAL STEP:** Distancing tubes from each other ensures that all samples heat uniformly and consistently, thus reducing the overall time required to reach the target temperature.

12.Once the metal rack has been filled, repeat steps 9–11 with a second rack. No space will be reserved for the control tube in this rack.13.Once all the tubes have been racked, spray the rack(s) with 70% IMS and transfer them to the oven, taking care to place the rack with the empty corner spot on the top shelf.14.Insert the thermometer probe into the control tube and place it in the reserved spot of the top rack.15.Close the oven and monitor the temperature of the control tube using the thermometer until it reaches 95 °C.
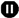
 **PAUSE STEP:** The internal sample temperature can take up to 23 min to reach 95 °C. Whilst a timer can be set for this time span, it is advisable to closely monitor the temperature, particularly when fewer than 200 samples are being inactivated.16.Once the control tube reaches 95 °C, the samples must remain in the oven for a further 5 min to complete inactivation.17.Remove the rack(s) from the oven using heat resistant gloves and transfer the sample tubes into the original barcoded 48-well storage racks.18.Place the storage racks at 4 °C to allow the samples to cool.
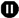
 **PAUSE STEP:** Inactivated samples can be stored safely at 4 °C for up to one week until processed.19.Repeat steps 9–18 for any remaining samples.20.Once transferred in barcoded storage racks, all the inactivated samples can be scanned using the FluidX Perception HD rack reader with the provided Intellicode software to obtain CSV files that can be imported into the information management software of choice.

### 3.3. Setting Up and Running qPCR Analyses on Heat-Inactivated Samples. Time for Completion: 2 h (for 90 Samples)

21.Determine the number of reactions to set up, including the control reactions, making sure to add extra reactions to account for pipetting error.22.Prepare an appropriate volume of primers/probe mix to add to the master reaction mix, according to Table 1.23.Prepare the required volume of master mix for the reactions to be set up, as indicated in Table 2.
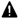
 **CRITICAL STEP:** UltraPlex 1-Step ToughMix has been specifically selected for this application due to its resilience to PCR inhibition caused by template impurities, a major concern when analysing saliva samples. The use of this specific mix is strongly recommended.

24.Dispense 12 µL of master mix into the wells of a 96-well PCR plate, including all the control wells.25.De-cap one or more racks of tubes containing the samples to be tested.26.Using the E1-ClipTip electronic adjustable multichannel pipette and 20 µL clip-tips, transfer 8 µL of saliva from the sample tubes to the appropriate wells of the qPCR plate.27.
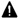
**CRITICAL STEP:** Adding an accurate amount of saliva is paramount to the correct outcome of the assay. Whilst heat treatment helps to reduce the intrinsic viscosity of the medium, thus allowing the vast majority of the samples to be transferred with satisfactory accuracy, specimens that do not comply with the submission requirements (e.g., provided after a meal or containing sputum rather than saliva) may occasionally prove problematic to pipette with the multichannel pipette. These samples will need to be transferred with a single-channel pipette, or the entire pipetting step will need to be repeated using the multichannel pipette. **NOTE:** We suggest that investigators explore chemical or enzymatic treatments to improve the consistency of saliva and the transferability of the samples.28.Re-cap the sample tubes.29.Add 8 µL of nuclease-free water or SARS-CoV-2-positive control sample to the non-template and positive control wells, respectively. To ensure the accuracy of the results and monitor the performance of the assay, the positive control should be loaded at double the viral particle concentration of the expected limit of detection for the assay being used (e.g., 1–2 viral particles/µL), as exemplified in Table 3.30.Seal the plate with transparent, PCR-grade adhesive film.31.Centrifuge the plate for 10–20 s to remove any bubbles generated during the pipetting steps.32.Once all bubbles have been removed, transfer the plate into the thermocycler.33.Set up and start a cycle as described in Table 4.

### 3.4. Data Analysis and Expected Results. Time for Completion: 5–15 min (for 90 Samples)

34.Once the PCR run is complete, export the data, and analyse the data with the software provided by the thermocycler manufacturer.
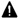
 **CRITICAL STEP:** The correct detection of each target (i.e., N, E, or RNase P) in the analysed sample will be indicated by the presence of sigmoidal amplification curves in the sample’s well, as visible in one (or more) fluorophore detection channels of the qPCR machine (i.e., FAM, HEX, Cy5). Furthermore, each generated curve will provide a threshold cycle (Ct) value that semi-quantitatively indicates the starting quantity of target in the sample (Figure 1).35.To confirm the purity of reagents and exclude the chance of widespread contamination, make sure that no amplification is detectable for any of the negative controls (Figure 2A).36.To confirm that the reaction setup and the assay have been performed correctly, make sure that N, E, and RNase P amplification is detectable for all the positive controls (Figure 2A).37.To confirm that all samples have been loaded correctly, make sure that RNase P amplification is detectable in all the loaded sample wells. Properly submitted samples that have been loaded optimally should yield amplification curves that cluster within a range of 5–10 Ct values and should also reach a plateau at similar RFU levels (Figure 2B).38.To determine the presence of SARS-CoV-2 within any sample, inspect all wells for N and/or E amplification curves. The detection of a sigmoidal curve for at least one target is indicative of the presence of virus within the sample (Figure 2C).
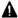
 **CRITICAL STEP:** It is strongly recommended to re-test samples that only yield amplification for one target, as to exclude the possibility of false positive results caused by a contamination event.39.To discern between a COVID-positive and -negative status for samples in which one or both SARS-CoV-2 targets have been detected, compare the Ct value(s) for each detected target against the cut-off value established for the assay.

## 4. Troubleshooting

### 4.1. Inconsistent or Unsuccessful Sample Loading

One of the major challenges when working with chemically untreated saliva is accurate and consistent pipetting, mostly due to the great variability in the viscosity and physical characteristics of samples submitted by different users. Whilst the assay is capable of tolerating fluctuations in the volume of sample added to the reaction mix, insufficient or excessive volumes can negatively affect target amplification by limiting the amount of available template or inhibiting the reaction itself, respectively (Figure 2B and Figure 3). Whilst a low RNase P curve could still be enough to validate a sample, the potential repercussions on N and E amplification could make the difference between calling a positive (albeit weak) and a negative result. Therefore, samples with unusually low Rnase P amplification must be repeated.

### 4.2. Contamination

The second most recurrent issue when analysing samples through qPCR is undoubtedly contamination. Saliva represents an even riskier analyte due to its viscosity, which facilitates cross-contamination of samples or reactions during pipetting operations. The aftermath of such events can be disrupting due to the increased risk of false positives. However, in most cases, contamination events can be distinguished from genuine results from the shape of the amplification curves that they generate (Figure 4). Specifically, contaminations are often assumed to provide linear plots (rather than sigmoidal amplification curves) due to the microvolumes transferred, resulting in poor amplification. Major contamination events, on the other hand, may yield plots that could be indistinguishable from those of genuine results, which is partly the reason why positive results are always validated by a second test. In general, potentially contaminated reactions should be readily repeated to confirm the result, although, in some rare occurrences where the contamination may have originated from (or spread to) the sample tube, the best course of action is procuring a second sample from the same donor or invalidating the result.

### 4.3. Sub-Optimal Amplification and Single-Target Amplification

Even in the absence of contamination and with optimal sample loading, a successful reaction can contain one or more samples that show poor or no amplification for one or both viral targets (Figure 5). Whilst not ideal, results of this kind are to be expected from samples that are not submitted correctly (e.g., over-diluted or altered by a drink) or otherwise contain a low viral load (i.e., obtained from donors at the beginning or end of a SARS-CoV-2 infection). Irrespective of the number of targets with poor or no amplification, the affected sample(s) should be re-tested to determine whether they are to be classified as positive, negative, or inconclusive (based on the Ct values obtained). Individuals who test inconclusive should be invited to provide an additional sample to confirm their COVID status.

### 4.4. Signal Drift

Suboptimal amplification and contamination can sometimes be confused with signal drift. However, the latter is an artifact that is usually characterised by an earlier appearance than most amplification curves and by a less steep slope than contamination curves that does not reach maximal amplification during the complete PCR cycle (Figure 6). Whilst this sort of distortion is not meaningful and can be automatically corrected by the analysis software, it is imperative that amplification curves from other samples have not been significantly altered by the algorithm. This is a risk during detection of low viral load or a contaminated sample and will therefore provide equivocal results.

## Figures and Tables

**Figure 1 mps-05-00025-f001:**
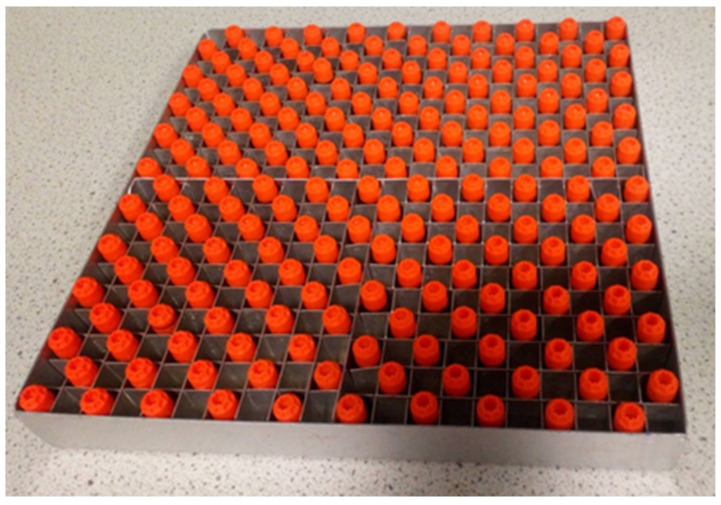
Correct loading of the metal oven rack for heat-inactivation.

**Figure 2 mps-05-00025-f002:**
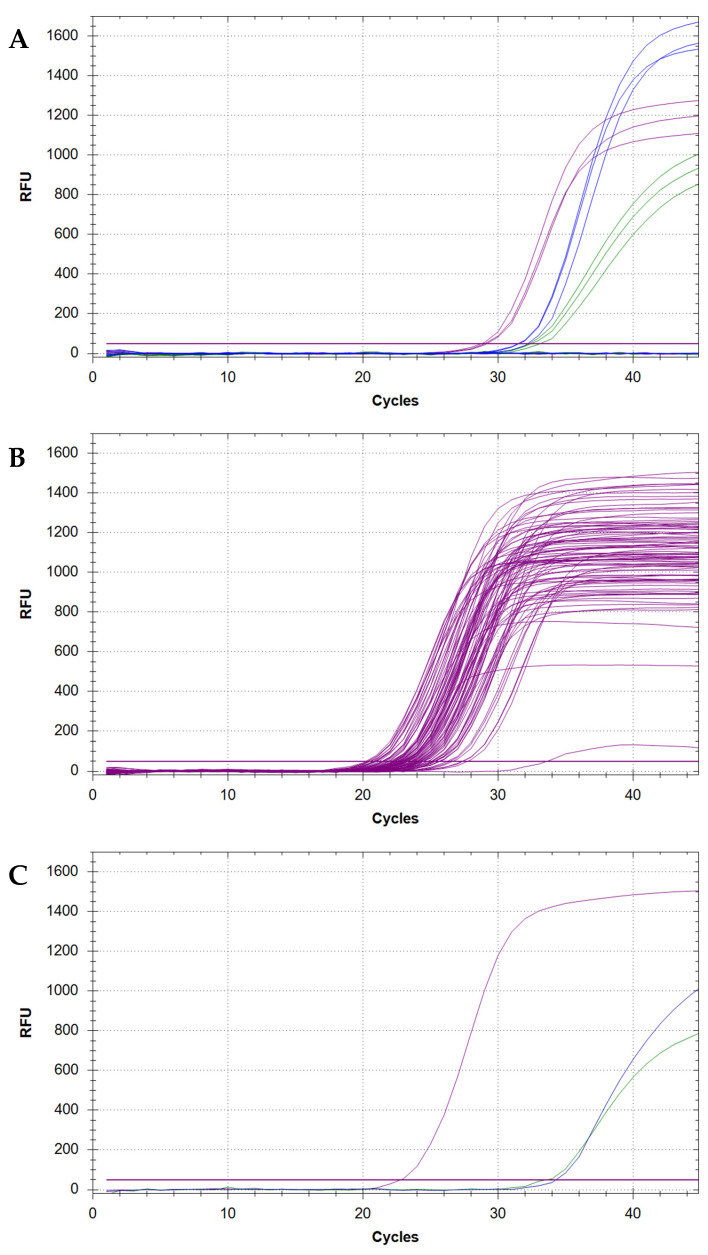
Examples of sigmoidal amplification curves obtained from heat-inactivated saliva samples positive or negative for SARS-CoV-2. (**A**) Amplification curves for N (green), E (blue), and RNase P (purple) genes as observed in three positive controls. Flat curves for the negative controls (no amplification) are visible at the bottom of the plot. (**B**) RNase P amplification curves for 90 samples analysed within the same PCR plate. Except from two outliers, all curves cluster together, yield Ct values between 20 and 28 and reach their plateau between 800 and 1500 RFU. (**C**) SARS-CoV-2 viral genome detected in a single sample as indicated by the presence of amplification curves for both N and E viral genes.

**Figure 3 mps-05-00025-f003:**
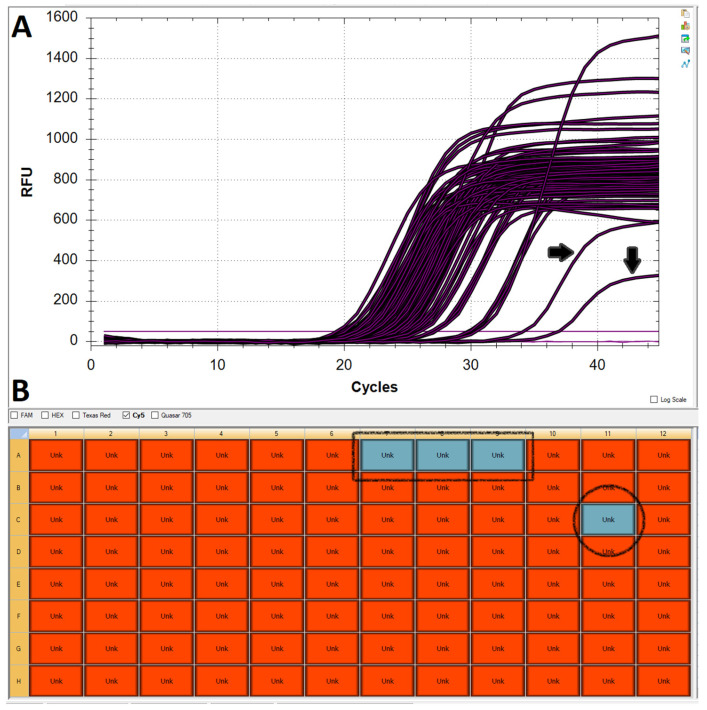
Effect of under- or over-pipetting saliva into the PCR reaction mix. (**A**) Two of the curves shown in the amplification plot for RNase P are shifted towards the right and have lower-than-average plateaus (black arrows), potentially due to the inhibitory effect of exceedingly high amounts of saliva in the reaction. (**B**) Red boxes indicate samples in the plate layout that yielded amplification for RNase P. Sample C11 (black circle) failed to produce an RNase P result, most likely due to the lack of template. Samples A7–9 (black square) are the no-template controls.

**Figure 4 mps-05-00025-f004:**
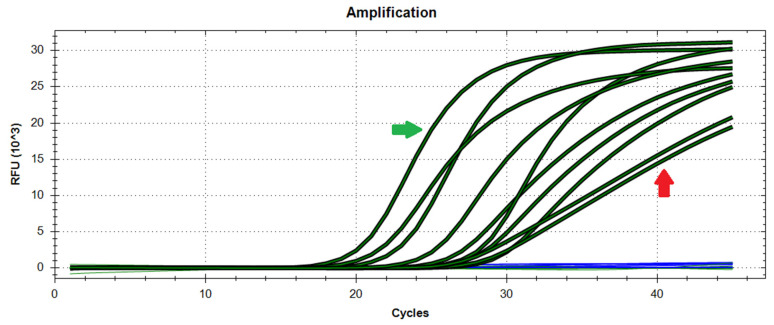
Difference between genuinely positive and contaminated samples. The curves to the left of the plot are from samples that are genuinely positive (green arrow), whereas two curves to the right are perfectly linear and suggest a contamination event (red arrow). Blue lines represent samples that have not generated amplification.

**Figure 5 mps-05-00025-f005:**
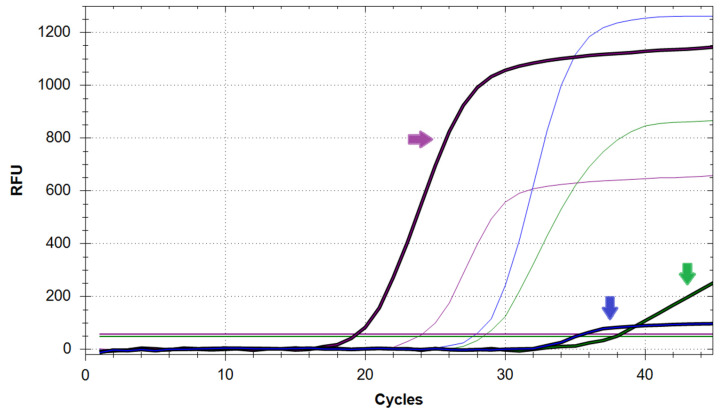
Saliva sample (bold curves) showing sub-optimal amplification of both N (green arrow) and E (blue arrow) genes but successful amplification of RNase P (purple arrow). Correct amplification of all three targets from the control sample is also shown (thin curves). The horizontal lines represent the RFU threshold for each target.

**Figure 6 mps-05-00025-f006:**
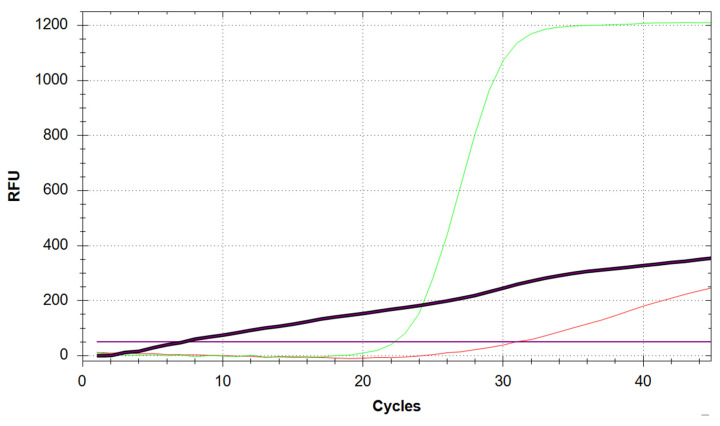
Comparison between genuine RNase P amplification (green curve), RNase P contamination (red curve), and signal drift (black curve) in the RNase P channel. The drift can be automatically corrected by the analysis software, but this could result in curves such as the red one also being flattened. The purple horizontal line represents the RFU threshold for RNase P.

**Table 1 mps-05-00025-t001:** Primers and probes required in the reaction mix. The final concentration of each reagent is indicated. A primers/probes mix Figure 100. µM stocks. Primers and probes required in the reaction mix. The final concentration of each reagent is indicated. A primers/probes mix for N reactions can be prepared in advance from 100 µM stocks.

Reagent	Final Concentration (nM)	Volume of Stocks (100 µM) for Primers/Probes mix (µL)
nCOV_N2_F	250	N × 0.05
nCOV_N2_R	250	N × 0.05
nCOV_N2_P (FAM)	62.5	N × 0.0125
E_Sarbeco_F1	200	N × 0.04
E_Sarbeco_R2	200	N × 0.04
E_Sarbeco_P1 (HEX)	100	N × 0.02
RNase P_F	62.5	N × 0.0125
RNase P_R	62.5	N × 0.0125
RNase P_P (ATTO647)	62.5	N × 0.0125

**Table 2 mps-05-00025-t002:** Reagents constituting the master reaction mix required for N individual tests.

Reagent	Volume for N Reactions (µL)
Nuclease-free water	N × 6.75
Primers/probes mix	N × 0.25
UltraPlex 1-Step ToughMix (4X)	N × 5.00

**Table 3 mps-05-00025-t003:** Reagents required for a 1.5 mL working stock of positive control containing 2 viral particles (vp) per microlitre.

Reagent	Volume (µL)
SARS-CoV-2 External Run Control (50 vp/µL)	60
SARS-CoV-2 Negative Control	200
Nuclease-free water	1240

**Table 4 mps-05-00025-t004:** Thermocycler settings for each step of the PCR cycle.

	Step	Description	Temperature	Time (min:s)
	1	Reverse transcription	50 °C	10:00
	2	Initial denaturation	95 °C	3:00
×45 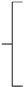	3	Denaturation	95 °C	0:03
4	Annealing	55 °C	0:30
5	Extension	72 °C	0:15

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
