# Peer review of "Direct RT-qPCR Assay for the Detection of SARS-CoV-2 in Saliva Samples"

_mps, 2022, doi:10.3390/mps5020025_

Round 1
Reviewer 1 Report
This paper presents a basic protocol for determining the viral load from saliva samples, which allows organizing mass testing using quantitative PCR analysis in order to increase the throughput of this method (processing more samples in the same time). This approach, in fact, is universal, and can be used not only when testing samples for the detection of SARS-CoV2 (since the pandemic seems to be on the decline), but also other viruses. No one canceled the possibility of a new serious pandemic, with possibly more severe consequences, when the need for mass testing of the population again arises. Of course, it will be necessary to make appropriate modifications with each new infection. The presented protocol is clear, each step is chosen meaningfully. Possible artifacts of the method are also analyzed. However, the reviewer has some questions that should be covered in this manuscript before publication.
1) CRITICAL STEP: Adding an accurate volume of saliva is paramount to the correct outcome of the assay. Samples that are difficult to pipette with the multichannel pipette will need to be transferred with a single channel pipette, or the entire pipetting step will need to be repeated using the multichannel pipette.
This is indeed a critical moment, and with a heavy load on the laboratory and uneven work of ordinary performers, it can lead to unreliable data. Is it possible to additionally affect saliva samples so that they become more or less evenly pipetted? Maybe an additional enzymatic treatment that does not affect the potential matrix will help?
2) The authors should present a comparative analysis of the time spent by the already widely used methods for determining the viral load with the protocol proposed in this paper. The speed of sample processing matters, sometimes even more important than differences in the accuracy of analysis by a few percent (if necessary, stop the development of the epidemiological process as soon as possible).
3) Line 233 etc. – Apparently, the authors made a mistake when numbering the figures. Here and below there is a reference to figure 1, while this should already be the second figure (the first one shows a photograph of the arrangement of test tubes). Accordingly, the numbers of subsequent figures should be changed.
Author Response
This paper presents a basic protocol for determining the viral load from saliva samples, which allows organizing mass testing using quantitative PCR analysis in order to increase the throughput of this method (processing more samples in the same time). This approach, in fact, is universal, and can be used not only when testing samples for the detection of SARS-CoV2 (since the pandemic seems to be on the decline), but also other viruses. No one canceled the possibility of a new serious pandemic, with possibly more severe consequences, when the need for mass testing of the population again arises. Of course, it will be necessary to make appropriate modifications with each new infection. The presented protocol is clear, each step is chosen meaningfully. Possible artifacts of the method are also analyzed. However, the reviewer has some questions that should be covered in this manuscript before publication.
- Thank you for your clear support and subsequent comments.
1) CRITICAL STEP: Adding an accurate volume of saliva is paramount to the correct outcome of the assay. Samples that are difficult to pipette with the multichannel pipette will need to be transferred with a single channel pipette, or the entire pipetting step will need to be repeated using the multichannel pipette.
This is indeed a critical moment, and with a heavy load on the laboratory and uneven work of ordinary performers, it can lead to unreliable data. Is it possible to additionally affect saliva samples so that they become more or less evenly pipetted? Maybe an additional enzymatic treatment that does not affect the potential matrix will help?
This is a valid point, that practically we have overcome by repeating the qPCR for that sample. However, discussion of possible ways to overcome this (albeit infrequent occurrence) are likely beyond the extent of this methods, but would like chang the paragraph to read “CRITICAL STEP: Adding an accurate amount of saliva is paramount to the correct outcome of the assay. Whilst heat treatment helps to reduce the intrinsic viscosity of the medium, thus allowing the vast majority of the samples to be transferred with satisfactory accuracy, specimens that do not comply with the submission requirements (e.g., provided after a meal or containing sputum rather than saliva) may occasionally prove problematic to pipette with the multichannel pipette. These samples will need to be transferred with a single-channel pipette, or the entire pipetting step will need to be repeated using the multichannel pipette.
NOTE: We suggest that investigators explore chemical or enzymatic treatments to improve the consistency of saliva and the transferability of the samples.
- 2) The authors should present a comparative analysis of the time spent by the already widely used methods for determining the viral load with the protocol proposed in this paper. The speed of sample processing matters, sometimes even more important than differences in the accuracy of analysis by a few percent (if necessary, stop the development of the epidemiological process as soon as possible).
- We agree in principle with this comment, but as we cannot perform a large scale comparison, we have altered the last paragraph of the introduction (around line 55) to include the following :
- Given the smaller scale of our operation, processivity has never been a limiting factor. However, the specificity and sensitivity demanded of the assay from the UK regulators were hard requirements that we had to meet. In practice, the occurrence of problematic samples is low enough to not have a meaningful impact on the overall speed of the process, and largely compensated by the time saved by not using other treatment options or extracting the RNA
3) Line 233 etc. – Apparently, the authors made a mistake when numbering the figures. Here and below there is a reference to figure 1, while this should already be the second figure (the first one shows a photograph of the arrangement of test tubes). Accordingly, the numbers of subsequent figures should be changed.
- We have made this amendment
Reviewer 2 Report
This manuscript by Tarantini et al. provides a detailed protocol for a direct RT-qPCR COVID-19 test from saliva samples. The assay employs a triplexed combination of DNA primer and probe oligonucleotides for the detection of the SARS-CoV-2 nucleocapsid (N) and envelope (E) genes, as well as the human RNase P gene as an internal control.
The manuscript is very clear and detailed. As such, I believe it should be published in Methods and Protocols. I have only some minor comments:
- A space is missing in line 39 before the parenthesis.
- In the legend of Fig. 2, samples in the black square are indicated as C7-9. However, based on the figure, I believe that they should be indicated as A7-9.
- Line 286-287. The manuscript reads: “However, in most cases, contamination events can be distinguished from genuine results from the morphology of the amplification curves that they generate (Figure 4)”. Regarding this point, I think that the reader would benefit from a few lines of explanation as to why contamination should give a linear plot rather than a sigmoidal amplification curve. If those contaminations are from SARS-CoV-2 material, shouldn’t they be amplified in a similar fashion? Are you referring to other types of contaminants? Please clarify this point by adding some lines of explanation.
Author Response
This manuscript by Tarantini et al. provides a detailed protocol for a direct RT-qPCR COVID-19 test from saliva samples. The assay employs a triplexed combination of DNA primer and probe oligonucleotides for the detection of the SARS-CoV-2 nucleocapsid (N) and envelope (E) genes, as well as the human RNase P gene as an internal control.
The manuscript is very clear and detailed. As such, I believe it should be published in Methods and Protocols. I have only some minor comments:
Response
Thank you for your approval, and for the clear revisions requested
- A space is missing in line 39 before the parenthesis.
- This has been amended
- In the legend of Fig. 2, samples in the black square are indicated as C7-9. However, based on the figure, I believe that they should be indicated as A7-9.
- This has been amended
- Line 286-287. The manuscript reads: “However, in most cases, contamination events can be distinguished from genuine results from the morphology of the amplification curves that they generate (Figure 4)”. Regarding this point, I think that the reader would benefit from a few lines of explanation as to why contamination should give a linear plot rather than a sigmoidal amplification curve. If those contaminations are from SARS-CoV-2 material, shouldn’t they be amplified in a similar fashion? Are you referring to other types of contaminants? Please clarify this point by adding some lines of explanation.
- We have added the additional text to explain this. “Specifically, contaminations are often assumed to provide linear plots (rather than sigmoidal amplification curves) due to the microvolumes transferred resulting in poor amplification. Major contamination events, on the other hand, may yield plots that could be indistinguishable from those of genuine results, which is partly the reason why positive results are always validated by a second test